# Olfactory Stimulation Successfully Modulates the Neurochemical, Biochemical and Behavioral Phenotypes of the Visceral Pain

**DOI:** 10.3390/molecules27217659

**Published:** 2022-11-07

**Authors:** Wen-Chieh Liao, Rou-An Yao, Li-You Chen, Ting-Yi Renn, Igor V. Klimenkov, Nikolay P. Sudakov, Fu-Der Mai, Yea-Tzy Chen, Hung-Ming Chang

**Affiliations:** 1Department of Post Baccalaureate Medicine, College of Medicine, National Chung Hsing University, Taichung 402202, Taiwan; 2Department of Anatomy and Cell Biology, School of Medicine, College of Medicine, Taipei Medical University, Taipei 110301, Taiwan; 3Department of Anatomy, School of Medicine, College of Medicine, Chung Shan Medical University, Taichung 402306, Taiwan; 4Graduate School of Biomedical and Health Sciences, Hiroshima University, Hiroshima 734-8553, Japan; 5Department of Cell Ultrastructure, Limnological Institute, Siberian Branch of the Russian Academy of Sciences, 664033 Irkutsk, Russia; 6Department of Biochemistry and Molecular Cell Biology, School of Medicine, College of Medicine, Taipei Medical University, Taipei 110301, Taiwan; 7Department of Speech Language Pathology and Audiology, College of Health Technology, National Taipei University of Nursing and Health Sciences, Taipei 112303, Taiwan

**Keywords:** central nucleus of amygdala, visceral pain, olfactory stimulation, neurochemical phenotypes, inflammation, time-of-flight secondary ion mass spectrometry (TOF-SIMS)

## Abstract

Visceral pain (VP) is the organ-derived nociception in which increased inflammatory reaction and exaggerated activation of the central nucleus of the amygdala (CeA) may contribute to this deficiency. Considering the amygdala also serves as the integration center for olfaction, the present study aimed to determine whether olfactory stimulation (OS) would effectively depress over-activation and inflammatory reaction in CeA, and successfully relieve VP-induced abnormalities. Adult rats subjected to intraperitoneal injection of acetic acid inhaled lavender essential oil for 2 or 4 h. The potential benefits of OS were determined by measuring the pro-inflammatory cytokine level, intracellular potassium and the upstream small-conductance calcium-activated potassium (SK) channel expression, together with detecting the stress transmitters that participated in the modulation of CeA activity. Results indicated that in VP rats, strong potassium intensity, reduced SK channel protein level, and increased corticotropin-releasing factor, c-fos, and substance P immuno-reactivities were detected in CeA. Enhanced CeA activation corresponded well with increased inflammatory reaction and decreased locomotion, respectively. However, in rats subjected to VP and received OS, all above parameters were significantly returned to normal levels with higher change detected in treating OS of 4h. As OS successfully depresses inflammation and CeA over-activation, application of OS may serve as an alternative and effective strategy to efficiently relieve VP-induced deficiency.

## 1. Introduction

Visceral pain is the nociceptive stimulation originated from activation of the nociceptors in the thoracic, pelvic, or abdominal organs. Most people have experienced visceral pain, and this disorder has become a significant social burden serving as a common cause for individuals to seek medical assistance [1]. However, although visceral pain represents a major clinical problem, yet far less is known about its central neuronal mechanism(s) [1,2]. Previous studies have indicated that enhanced activation of the central nucleus of the amygdala (CeA) may play an essential role in the formation of visceral pain-induced sensory and behavioral responses [3,4,5]. Pharmacological reports also demonstrated that over-expression of the stress transmitters [e.g., corticotropin-releasing factor (CRF) and substance P (SP)], together with reduced activation of the small-conductance calcium-activated potassium channels (SK channels) in the CeA may act as the neurochemical substrate contributing to the increment of neuronal activity and facilitation of the synaptic transmission of pain [6,7,8,9,10,11]. Moreover, changes in the microenvironment associated with physiological distress would activate the microglia that could further modify the neuronal activities of CRF neurons by potentiating the synaptic remodeling and releasing pro-inflammatory cytokines [12,13,14]. Through mediating the medium type after-hyperpolarization (mAHP), and favoring the *N*-methyl-D-aspartate (NMDA) receptor phosphorylation through the recruitment of protein kinase A (PKA), activated neurochemical modulation would increase the excitatory drive of CeA neurons and potentiate the nociceptive signaling [15,16,17]. As hyperactivity of CeA and excessive activation of microglia may underlie the neuronal mechanism of visceral nociception, delivering agent(s) that directly target the CeA and effectively depress the amygdaloidal activity might be worthy of trial for clinical use as a novel therapeutic strategy to relieve the physiological distress and improved the quality-of-life for those suffering from visceral pain [12,18].

Lavender essential oil (LEO) is the natural product of phytochemicals extracted from the genus *Lavandula*, which has been well known for its wide use in perfumes, cosmetics, food processing and massage for many centuries [19]. During the past few years, the potential use of LEO in medical applications has attracted numerous attention due to its beneficial effects on relieving a variety of disorders [20,21]. Previous studies have indicated that LEO possesses significant antibacterial, antifungal, carminative, sedative, antidepressant, and antinociceptive properties [22,23,24,25,26]. Biochemical reports also revealed that LEO inhalation could reduce c-fos expression in hypothalamic nuclei and CeA, which effectively controls the downstream expression of genes expected to be involved in certain situations [23,27]. It is indicated that the olfactory stimulation originated from nasal inhalation would firstly project to the main olfactory bulbs and then reach the CeA through the dense intra-amygdaloidal neuronal circuits [28,29].With regard to the above viewpoint that amygdala serves as an important structure integrating both olfactory and nociceptive stimuli, it is thus reasonable to suggest that inhaling LEO may exclusively target the CeA, and therefore relieves the visceral nociception through depressing the neurochemical profiles of the CeA neurons.

However, although the functional role of CeA in the modulation of visceral pain is well documented, the spatiotemporal expression of its related neurochemicals (i.e., c-fos, CRF, SP, and SK channels) and the extent of microglial expression involved in the activation of CeA following visceral nociception have never been systemically reported. Moreover, whether olfactory stimulation (OS) via the inhalation of LEO would effectively correct the neurochemical expression, depress microglia activation, and successfully relieve the biochemical and behavioral phenotype of the visceral pain is still remained to be explored. As attempts to answer these questions, the present study was firstly aimed to quantitatively examine the neurochemical and microglial expression in the entire region of the CeA following the irritation of visceral pain. Secondly, in order to determine whether the application of OS would exert substantial benefits on depressing the visceral pain, the potential changes of the neurochemical and microglial activation in response to LEO were extensively detected by spectrometric, biochemical, and morphological analysis, respectively. Thirdly, with the aim of providing the functional evidence of OS on relieving the visceral pain, the performance of locomotor activities such as abdominal writhing, frequency of rearing behavior, and number of grooming movements were further assessed in the current study. Finally, for the purpose of evaluating the stressful extent accompanied with the suffering resulted from visceral pain, the urine level of norepinephrine (NE) was measured as the biochemical marker of stress experienced by the animals. The experimental model of visceral pain was achieved by intraperitoneal injection of acetic acid, since this paradigm represents the slow form of chemical stimulation that could produce an inescapable and stereotyped behavior closest in nature to clinical pain [30]. By using this model, our previous study has successfully reported that a prolonged feeling of pain would be induced and suffered, which therefore provides a sufficient time frame for OS to exert its maximal effects [31]. Nevertheless, although this model is widely used as a visceral pain model, it would probably be more proper to call it peritoneal–visceral pain since the intraperitoneal injection of algogenic agents would also irritate the parietal peritoneum that receives somatic innervation [32].

## 2. Results

### 2.1. OS Effectively Suppresses the *CRF* Expression in the CeA Following Visceral Pain

In rats subjected to intraperitoneal injections of saline and exposed to either water vapor or LEO, nearly no CRF immuno-reactive neurons were detected in the CeA (Figure 1A). In contrast, in rats subjected to intraperitoneal injection of 2% acetic acid and who received water vapor, numerous CRF immuno-reactive neurons were detected in the CeA (arrows, Figure 1B) with preferential localization to the rostral part of this nucleus. Nevertheless, in rats subjected to the induction of visceral pain and who received LEO inhalation, the CRF immuno-reactive neurons were drastically decreased, in which just a few neurons were slightly labeled by CRF immunohistochemistry (Figure 1C,D). Quantitative evaluation coincided well with immunohistochemical findings in which the number of CRF immuno-reactive neurons was greatly reduced from 43 ± 10 to 6 ± 3 in the visceral pain-treated group and OS group, respectively.

### 2.2. OS Significantly Suppresses the SP Expression in the CeA Following Visceral Pain

In rats subjected to intraperitoneal injection of saline and exposed to either water vapor or LEO, only a few SP immuno-reactive neurons were detected in the CeA (Figure 2A). In contrast, in rats subjected to intraperitoneal injection of 2% acetic acid and who received water vapor, extensive increases of SP immuno-reactive neurons were detected in the entire region of CeA with no preferential localization (Figure 2B). However, in rats subjected to visceral pain and who received LEO inhalation, the number of SP immuno-reactive neurons in the CeA was significantly decreased with the higher reduction detected in the group that inhaled LEO for 4 h (Figure 2D). The reduction of SP expression following LEO inhalation was also demonstrated by quantitative evaluation in which the number of SP immuno-reactive neurons was counted to be 55 ± 5 in the visceral pain group and 13 ± 3 in the OS group (LEO 4 h), respectively.

### 2.3. OS Effectively Decreases the c-fos Expression in the CeA Following Visceral Pain

There were only a small number of c-fos immuno-reactive nuclei detected in the CeA (25 ± 3/500 μm^2^) following intraperitoneal injection of saline and exposure to water vapor or LEO (Figure 3A). However, in rats subjected to intraperitoneal injection of 2% acetic acid, a large number (73 ± 15/500 μm^2^) of c-fos immuno-reactive nuclei were detected in the CeA (Figure 3B). Nevertheless, in rats subjected to visceral pain and who received LEO inhalation for 2 h and 4 h, the number of c-fos immuno-reactive nuclei was significantly decreased (Figure 3C,D). As c-fos is a reliable marker for neuronal activation [33], reduction of c-fos expression following OS indicates that LEO inhalation can effectively depress the neuronal activity of CeA, and therefore, relieves the excitatory drive of CeA in response to visceral pain.

### 2.4. OS Successfully Increases the SK Channel Protein Level in the CeA Following Visceral Pain

In rats subjected to intraperitoneal injection of saline and exposed to either water vapor or LEO, significantly high levels of SK channel protein were detected in the CeA (Figure 4). In contrast, in rats subjected to intraperitoneal injection of 2% acetic acid and who received water vapor, a considerable decrease of SK channel protein level was detected in the CeA (Figure 4). Nevertheless, in rats subjected to the induction of visceral pain and who received LEO inhalation, the SK channel protein level was effectively increased in the CeA, in which no significant difference in the protein level of SK channel was observed between OS and untreated groups (Figure 4). As the SK channel plays an important role in regulating the neuronal activity of CeA, increased expression of the SK channel protein following OS may have potential benefits in antinociceptive processes after visceral pain.

### 2.5. OS Successfully Normalizes the Intracellular K^+^ Level in the CeA Following Visceral Pain

In rats subjected to intraperitoneal injection of saline and exposed to either water vapor or LEO, only a moderate K^+^ signal was detected in the CeA (Figure 5A). Most of the K^+^ signal was distributed in the intracellular portion of the CeA neurons (arrows, Figure 5B). However, in rats subjected to visceral pain and who received water vapor, an intense K^+^ level was detected in the CeA (Figure 5D) in which a strong K^+^ signal was clearly observed in the intracellular portion of the CeA neurons (arrows, Figure 5E). The extensive high levels of the intracellular K^+^ (Figure 5E) paralleled well the drastic reduction of the SK channel protein level (Figure 4), which mediates the outflow of K^+^ and increases the excitatory drive of CeA neurons in response to nociceptive stimuli. Nevertheless, in rats subjected to visceral pain and who received LEO inhalation, the K^+^ signal of the CeA was significantly returned to nearly normal level as revealed by both terms of spectral intensity (Figure 5G) and ionic imaging (Figure 5H). Quantitative evaluation revealed that the normalized spectral intensity of the K^+^ signal following OS treatment is counted to be 380 ± 107%, much less than that receiving the visceral pain (787 ± 46%) (Figure 5J).

### 2.6. OS Significantly Depresses Microglia Activation and Reduces Pro-Inflammatory Cytokine Level in the CeA Following Visceral Pain

By using the Iba1 immunohistochemistry, a few microglia with a distinct ramified profile were detected in the CeA of untreated rats (arrow, Figure 6A,E). However, following visceral pain, enhanced microglial activation with an amoeboid profile characterized by thickened and retracted branches along with enlarged cell bodies was observed in the CeA (arrow, Figure 6B,F). Nevertheless, in rats subjected to visceral pain and who received LEO for 2 or 4 h, both the number and morphology of the microglia was gradually returned to untreated levels in which the better recovery was detected in the animals inhaling LEO for 4 h. (Figure 6C,D,G). Morphometric analysis of the branching type of microglia corresponded well with the immunohistochemical findings in which a significantly lower D value (i.e., amoeboid type) was calculated in the visceral pain group as compared to that of untreated and OS ones. Moreover, the pro-inflammatory cytokine level was in good agreement with the microglial expression, in which increased levels of TNF-α, IL-1β, and IL-6 were detected in the animals subjected to visceral pain (Figure 7) which were also highly expressed in the activated microglia (Figure 6B). In addition, it is also worthy to note that in animals subjected to visceral pain and who received LEO, the level of pro-inflammatory cytokines was significantly decreased as a consequence of reduced microglia activation (Figure 6D and Figure 7).

### 2.7. OS Significantly Declines the Urinary NE Level Induced by Visceral Pain

In the present study, the urinary NE level was slightly elevated in rats subjected to intraperitoneal injection of saline and exposed to either water vapor or LEO (Figure 8). However, in rats subjected to visceral pain, the urinary NE level was drastically increased to nearly five-fold (159 ± 23 μg/L) that of the untreated ones (31 ± 5 μg/L) (Figure 8). Nevertheless, in rats subjected to visceral pain and exposed to LEO inhalation, the stress level was successfully decreased in which a higher reduction was observed in rats receiving LEO for 4 h (38 ± 2 μg/L) (Figure 8).

### 2.8. OS Successfully Improves the Behavioral Responses Following Visceral Pain

As shown in Figure 9, significant behavioral changes such as increased numbers of abdominal writhing (Figure 9A), decreased rearing behavior (Figure 9B), and reduced grooming movements (Figure 9C) were observed in the rats subjected to visceral pain. However, in rats subjected to visceral pain and who received LEO inhalation, all the above behaviors were gradually diminished, in which less abdominal constriction, high frequency of rearing behavior, and an increase in the number of grooming movements were all detected in the animals within the 4 h of inhaling period. Due to the fact that abdominal writhing is the distress feature of severe visceral pain, relieving the occurrence of abdominal writhing after LEO application (Figure 9A) suggested that LEO may exert significant anti-nociceptive effects probably through depressing the neuronal activity and reducing the extent of microglial activation within the CeA.

## 3. Discussion

The present study has provided the first functional anatomical evidence that application of OS by continuous inhalation of LEO could successfully modulate the neurochemical (Figure 1, Figure 2 and Figure 3), biochemical (Figure 4, Figure 5, Figure 6, Figure 7 and Figure 8), and behavioral (Figure 9) phenotypes of the visceral pain. By reducing the CRF, SP, and c-fos expression (Figure 1, Figure 2 and Figure 3), increasing the SK channel protein level (therefore normalize the trans-membranous ion gradient) (Figure 4 and Figure 5), depressing the extent of microglia activation and the corresponding level of pro-inflammatory cytokines (Figure 6 and Figure 7), LEO could suppress the excessive neuronal activity and inflammatory state of CeA, which consequently exert an anti-nociceptive effect on the modulation of pain. The beneficial effect of LEO on relieving the visceral pain is time-dependent, as the better effect was detected at 4 h following the exposure to LEO. It is indicated that the CeA is the central structure responsible for the modulation of sensory and behavioral responses associated with the visceral pain [3,4,5]. Nociceptive inputs from the ascending spinoparabrachial tract and collateral branching of the spinothalamic tract, as well as directly from the dorsal horn of the spinal cord, would reach the CeA, and contribute to the neurochemical alterations of the CeA in response to the painful stimuli [29,34]. It has been reported that enhanced expression of CRF in the CeA would activate the neuronal circuitry that is engaged in facilitating the synaptic transmission of pain [8.9]. Pharmacological studies also demonstrated that up-regulation of SP in the CeA may be involved in stress regulation and may play an important role in nociceptive integrations [6,7,35,36]. Through over-activated CRF-ergic and SP-ergic projections, enhanced drive of the CeA could influence widespread regions of the basal forebrain and periaqueductal gray matter (PAG), which directly and indirectly projects to the brainstem and hypothalamus, known to be participated in the triggering of autonomic and behavioral responses [18,35,36,37,38]. The present findings thus coincided well with these viewpoints in which we successfully detected an increased expression of CRF and SP in the CeA following the induction of visceral pain (Figure 1 and Figure 2). Increased expression of these neurochemicals paralleled well with the incidence of autonomic and behavioral impairments as demonstrated by extensive high level of NE (Figure 8) and impaired locomotor activities (Figure 9). In addition, the hyper-excitation of CeA induced by visceral pain was further demonstrated by the intensive labeling of c-fos (a reliable marker of neuronal activation [33]) and decreased activation of the SK channel, which up-regulates the neuronal excitation of CeA neurons by increasing the firing frequency of action potentials and promoting synaptic plasticity [10,39] (Figure 3, Figure 4 and Figure 5). Based on these findings that hyperactivity of the CRF and SP systems in the CeA may serve as the neurochemical substrate contributing to the development of autonomic or behavioral impairments induced by visceral pain, delivering agent(s) that could directly target the CeA and depress CeA hyperactivation might be worthy of trial for clinical use as a novel strategy to relieve the physiological distress resulted from visceral pain. This is just the case that in our current study, we have clearly demonstrated that inhalation of LEO would exert significant effects on suppressing the CRF and SP expression in the CeA (Figure 1 and Figure 2). Reduction of both neurochemicals was positively correlated with the depression of neuronal excitation (Figure 3, Figure 4 and Figure 5), decrease in stress levels (Figure 8), and a considerable improvement of behavioral activities (Figure 9). To the best of our knowledge, this study is the first one using spectrometric, biochemical, and quantitative morphological approaches to systemically reveal the neurochemical mechanisms regarding the effectiveness of LEO on the relief of visceral pain. Considering that chronic visceral pain still remains highly prevalent and poorly treated due to its complex pathophysiology, the application of OS via inhaling LEO may be used as an alternative but effective strategy to counteract the suffering induced by visceral pain through an easier, safer, more economic, and more convenient way.

On the other hand, the present study also highlights the functional role of enhanced microglial activation within the CeA on the modulation of visceral pain. It is well known that microglia is the innate immune cell in the central nervous system that participate in the maintenance of neuronal homeostasis in response to microenvironment changes such as immune challenges or nociceptive stimuli [40]. Due to their highly plastic characteristics, activated microglia would undergo structural and functional changes that may point to the critical involvement in the central mediation of visceral pain [12]. Previous studies have indicated that microglia in the CeA and limbic system could regulate visceral nociception and lead to the occurrence of stress-induced anxiety and depressive-like behaviors [12,13,14,41]. Pharmacological reports also demonstrated that microinfusion of minocycline, a microglial inhibitor, into the CeA would directly reverse the morphological changes of the microglia, prevent the microglia-mediated synaptic engulfment, and effectively depress the stress-induced visceral pain [42]. Through modifying the synaptic remodeling and release of pro-inflammatory cytokines, activated microglia could intensify the neuronal activity of CeA that consequently contributes to the induction of visceral hypersensitivity [12,14]. Our current study was in good agreements with these findings in which enhanced microglia activation (Figure 6), together with increased level of pro-inflammatory cytokines (Figure 7), was clearly detected in the CeA of adult rats subjected to visceral pain. Activated microglia also correlated well with the reduced expression of the SK channel (Figure 4) which further elicited an excessive excitatory drive of CeA neurons and aggravated the microglia-mediated inflammatory activity [10,11,43]. As reduced expression of SK channel in CeA plays an important role in enhancing microglia activation and nociceptive transmission [10,11,43], increasing SK channel levels by LEO (as demonstrated in our present study, Figure 4) strongly indicates that LEO would exert a significant anti-nociceptive effect following visceral pain possibly through both directly depressing the neuronal excitability of CeA (Figure 1, Figure 2 and Figure 3) and indirectly inhibiting the microglia-mediated inflammatory activities (Figure 6 and Figure 7).

However, although the use of LEO in medical applications is increasingly popular in many countries during the past few decades [19], the overall compounds of LEO are still not fully understood. The biological functions of LEO are generally supposed to be resulted from its major constituents such as linalool, linalyl acetate, and terpenoids [44,45]. A previous study has indicated that linalool and its ester, linalyl acetate, would exert a significant anti-inflammatory effect on the carrageenin-induced rat edema model [46]. Pharmacological reports also demonstrated that linalool and terpenoids could prevent nociception in different models originated from thermal, chemical, and inflammatory insults [47,48,49]. By effectively modulating the expression of related neurochemical and receptors in several neuro-circuitries (such as cholinergic, glutamatergic, and dopaminergic systems), linalool and terpenoids can alter the activities of specific brain regions, which thereafter block the processes of nociceptive neurotransmission [50,51,52,53]. Our current study thus clearly supported these findings in which we also detected a decreased CRF and SP expression in the CeA following LEO inhalation (Figure 1 and Figure 2). Decreased expression of these neurochemicals after LEO corresponded well with the reduction of c-fos and CeA excitation (Figure 3, Figure 4 and Figure 5), depression in microglia activation and pro-inflammatory cytokine production (Figure 6 and Figure 7), decline of stress level (Figure 8), and ultimately contributed to the improvement of locomotor activities (Figure 9). As the intranasal administration provides a distinct route allowing numerous molecules to enter the central nervous system without being excluded by the blood–brain barrier [54,55], effective compounds in the LEO may directly target the CeA through the olfactory pathway, and traverse the cell membranes to exert their powerful anti-nociceptive activity.

In addition to acting as an anti-nociceptive agent, the sedative function and psychological effects of LEO on nociceptive responses should not be overlooked. It has been reported that the sedative action of an essential oil would greatly drop the behavioral motility, which may serve as a confounding factor for anti-nociceptive functions [56,57]. Pharmacological reports also demonstrated that specific pleasant odors released from LEO may possibly be associated with positive mood and emotions, which may further take part in the modulation of discomfort behavior or relief of nociceptive memory [58]. In our current study, no decrease in spontaneous locomotor activities was observed in the animals subjected to LEO inhalation (data not shown), suggesting that the sedative action of LEO would not affect LEO’s antinociceptive efficacy. On the other hand, it must be noted that the current model used in this study is a mixed visceral and somatic one, because the parietal peritoneum, which is a partly somatic structure, is also involved [32]. With regard to this viewpoint, any nociceptive information originated from visceral organs may be associated, or modulated by inputs coming from somatic ones [32]. Based on this reason, it is somewhat difficult to clearly identify and differentiate whether, and to what extent, the neurochemical and the behavioral profiles were provoked by the visceral or by the somatic structures [59]. Considering that the amygdala may present divergent effects on regulating the visceral and somatic pain [60,61], comprehensively comparing the responses induced by current study with those stimulated by real somatic irritant would provide further insight into the potential mechanisms underlying the antinociceptive function of LEO against the noxious stimuli arisen from different origins. Furthermore, it is also worthy to mention that in recent years, a growing body of evidence has shown that both gender and age would play a significant role on modulating the neuroanatomical circuits and physiological processes of visceral pain [62,63,64,65]. Significant sex and age differences in neuronal morphology, synaptic organization, cell proliferation, and microglial activity have clearly been demonstrated in the amygdala and several corticolimbic regions [64]. Moreover, greater activation of the amygdala in response to visceral pain has also been reported in female rats in which females showed activation in broad areas of the amygdala, while males only activated a more circumscribed region of this nucleus following visceral pain [63]. Although the present study aimed to illustrate the beneficial effects of LEO in a more straightforward way, the importance of including female animals, as well as delineating the potential changes in nociceptive processing across different ages should be further highlighted in future studies. Finally, it is interesting to point out that a hemispheric lateralization of nociceptive function has also been demonstrated in the amygdala in which the right CeA appears largely more implicated in the perception of pain than that of the left one [66,67]. Although we did not find any difference in neurochemical and biochemical profiles (at least for CRF, SP, c-fos, SK channel, and microglia expression) between these two sides of the CeA following visceral pain and LEO inhalation, the potential anatomical and functional basis for this lateralization still remains an important issue worthy of further investigation in future studies.

## 4. Materials and Methods

### 4.1. Treatment of Experimental Animals

Adult male Wistar rats (*n* = 36, weighing 230~250 g) obtained from the National Laboratory Animal Center were used in this study. The experimental animals were divided into two groups equally. Rats in the first group were intraperitoneally injected with 1 mL of saline while those in the second group were subjected to intraperitoneal injection of 2% acetic acid (5 mL/kg) [30]. Immediately after the intraperitoneal injection, both groups were subdivided into three subgroups with *n* = 6 in each. Animals in these subgroups were then placed in the evaporation chamber (with Wide × Length × Height = 39 cm × 33 cm × 25 cm) and inhaled either water vapors or LEO (200 μL, No.61718, Sigma, St. Louis, MO, USA) via an ultrasonic diffuser placed at the bottom of the chambers for 2 or 4 h. All the experiments were performed between 8:00–12:00 h. During the entire experimental period, all rats were exposed to an automatically regulated light–dark cycle of 12:12 (light on 7:00–19:00h) at a constant temperature of 25 °C and relative humidity of 60%. The animals were allowed to food and water ad libitum. In the care and handling of all experimental animals, the Guide for the Care and Use of Laboratory Animals (1985) as stated in the United States NIH guidelines (NIH publication No. 86–23) was followed. All the experimental procedures and drug administration processes were further approved by the Laboratory Animal Center Authorities of the Taipei Medical University.

### 4.2. Observation and Quantification of Behavioral Responses

In order to test the physiological distress of acetic acid-induced visceral pain, as well as the potential benefits of LEO inhalation on the alteration of behavioral responses, writhing numbers of the abdominal musculature, rearing behavior, and grooming movements were used as the indicators for behavioral analysis. The number of writhing movements (such as abdominal constriction, arching of back and trunk, and twisting or stretching of hind limbs) was cumulatively counted and recorded by a video camera suspended above the evaporation chamber for 4 h. For the measurement of rearing and grooming behaviors, the number of rearing events, as characterized by an action of keeping body in a vertical state after making the legs straight and lifting buttocks of the floor, as well as the number of an action of lifting hands to ears, was also recorded and calculated for 4 h. All behavioral tests were recorded and counted by an experimenter not involved in the rest of the experiments. All data were analyzed subsequently in a double-blind fashion.

### 4.3. Perfusion and Tissue Preparation

For performing a quantitative immunohistochemical study and time-of-flight secondary ion mass spectrometry (TOF-SIMS) analysis unbiasedly, half the amount of the animals in each experimental subgroup (*n* = 18) were randomly selected and deeply anesthetized with xylazine (10 mg/kg) and ketamine (100 mg/kg) and perfused transcardially with 0.9% saline followed by 300 mL of 4% paraformaldehyde in 0.1 M phosphate buffer (PB), pH 7.4. After perfusion, the forebrain segment containing the amygdala was removed and kept in the same fixative for 2 h. The tissue block was then immersed in graded concentrations of sucrose buffer (10~30%) for cryoprotection at 4 °C overnight. Serial 30-μm-thick sections of the amygdala were cut transversely with a cryostat (CM3050S, Leica Microsystems, Wetzlar, Germany) on the following day and were alternatively placed into six wells of a cell culture plate. Sections collected in the first four wells were subsequently processed for quantitative CRF, SP, c-fos, and Iba1 immunohistochemistry, respectively, while those in the fifth well were attached to silica wafers and processed for TOF-SIMS analysis. For those sections collected in the last well, they were processed for regular hematoxylin and eosin (H&E) staining.

### 4.4. Corticotropin-Releasing Factor, Substance P, c-fos, and Iba1 Immunohistochemistry

For CRF, SP, c-fos, and Iba1 immunohistochemistry, the forebrain sections containing the amygdala were first washed in three changes of 0.01 M phosphate-buffered saline (PBS), pH 7.4, and then placed in 0.01 M PBS containing 3% hydrogen peroxide for 1 h to reduce the endogenous peroxidase activity. Following three rinses in PBS, sections were incubated in the blocking medium containing 0.1% Triton X-100, 3% normal goat serum and 2% bovine serum albumin (all from Sigma, St. Louis, MO, USA) for 1 h to block nonspecific binding. After several washes in PBS, the sections were incubated with the respective primary antibodies (CRF: 1:1000, substrance P: 1:100, c-fos: 1:3000, Iba1: 1:1000, all from Abcam, Cambridge, UK) in blocking medium at 4 °C for 24 h. After incubation in primary antibodies, the sections were further incubated with a biotinylated secondary antibody (1:200) (Vector Laboratories, Burlingame, CA, USA) at room temperature for 2 h, followed by the standard avidin–biotin complex procedure (Vector Laboratories, Burlingame, CA, USA) with diaminobenzidine as a substrate of peroxidase. Before photomicrographic studies, all reacted sections were mounted on gelatinized slides, dehydrated through a graded series of alcohol, cleared with xylene, and coverslipped with Permount.

### 4.5. TOF-SIMS Analysis

TOF-SIMS analysis was carried out on a TOF-SIMS IV instrument (ION-TOF GmbH, Münster, Germany) as described in our previous studies [68,69]. Fixed cryostat sections were attached to silica wafers and the temperature of sample holder was adjusted to −60 °C. A bismuth (Bi^+^) ion gun operated at 25 kV was used as the primary ion source (1 pA pulse current). The Bi^+^ primary ion beam was scanned over an area of 500 µm^2^ which included 128 × 128 pixels. Image data acquisition time was 200 s with the best resolution obtained at m/Δm = 7450. Positive secondary ions flying through a reflectron mass spectrometer (ION-TOF GmbH, Münster, Germany) were detected with a micro-channel plate assembly operating at 10 kV post-acceleration. The paraformaldehyde and a set of standard peaks [like *m*/*z* 15 (CH_3_^+^), 27 (C_2_H_3_^+^), and 41 (C_3_H_5_^+^)] were used as a mass calibration to ameliorate the potential matrix effect for ion spectrums [70]. The ions related to *m*/*z* 39.09 were used to identify and evaluate the ionic image of K^+^.

### 4.6. Immunobloting Assay for SK Channel Protein Level

For immunoblotting assays, another half amount of the animals in each experimental subgroup (*n* = 18) were decapitated under deep anesthesia with xylazine (10 mg/kg) and ketamine (100 mg/kg). After removal of the skull, the brain segments containing the amygdala were quickly removed and bisected equally throughout the midline. Following that, one side of the bisected segment was processed for pro-inflammatory cytokine measurements (please see below), while the other side was frozen in liquid nitrogen and then homogenized with 100 μL lysis buffer using a grinder on ice. The immunoblot procedure was processed by the methods describe previously [64]. Briefly, 10 μg of solubilized proteins were separated on SDS-PAGE (12%) and electroblotted onto a polyvinylidene difluoride (PVDF) membrane (Bio-Rad Laboratories, Hercules, CA, USA). The membranes were blocked with 5 % non-fat dry milk and probed sequentially with antibodies against β-actin (1:5000) and the SK channel (KCNN3, 1:1000). Following that, the PVDF sheets were incubated with HRP-conjugated secondary antibody at a dilution of 1:5000 for 1 h at room temperature. The immunoreaction was visualized with ECL solution (5 min) followed by 2 min of film exposure. Optical density (OD) was quantified with a computer-assisted software (Science Lab 2003, Fuji, Japan). Densitometric results were normalized against β-actin and are presented as mean ± standard deviation (SD).

### 4.7. Luminex Analysis of the Pro-Inflammatory Cytokine Levels

For measuring the pro-inflammatory cytokine levels, the total protein was extracted from the isolated CeA of the bisected segment (*n* = 18, as described above) using the Bio-Plex cell lysis kit (Bio-Rad Laboratories, Hercules, CA, USA). After that, the total protein concentrations were determined with the Pierce BCA protein assay kit (Thermo Fisher, Waltham, MA, USA). The pro-inflammatory cytokine levels were then measured using the Bio-Plex Pro rat cytokine 23-plex assay kit on the Bio-Plex 200 system following the manufacturer’s instructions. The results are expressed as the mean pg/mg total protein ± SD of the concentrations of each factor.

### 4.8. Biochemical Measurement of Urinary Level of Norepinephrine

As norepinephrine (NE) is released from the nervous system and adrenal gland after autonomic excitation induced by harmful stimuli, elevated levels of urinary NE may serve as a useful biochemical marker indicating the stress level experienced by the animals. During the 2 or 4 h period of water vapor or LEO inhalation, the urine samples of all experimental animals (*n* = 36) were collected and processed for urinary levels of NE by the commercial kit (KA 1891, Abnova, Taipei, Taiwan) according to the manufacture’s guidelines.

### 4.9. Quantitative Study and Image Analysis for Neurochemical Phenotypes

The computerized image analysis for quantifying the expression pattern of neurochemicals was carried out in those animals subjected to immunohistochemistry (*n* = 18). The general approach for quantitative image analysis was similar to our previous studies [68,71]. A computer based image analysis system along with the Image-Pro Plus software 6.0 (Media Cybernetics, Silver Spring, MD, USA) was used to quantify the amount of positive labeling. A digital camera mounted on a ZEISS microscope (Axioplane 2, Carl Zeiss MicroImaging GmbH, Hamburg, Germany) imaged sections and displayed them on a high-resolution monitor. A total of 20 sections per animal, representing the entire rostro-caudal extent of the CeA, were analyzed. The number of CeA neurons (or nuclei) reacted for CRF, SP, and c-fos was then densitometrically counted, and all readings were combined and averaged to obtain the total amount of labeling. For quantifying the morphological changes of the microglia (ranging from simple rounded to complex branched), the value of fractal dimension (D) was used as an indicator in which a higher D means a greater complexity of the expression pattern [72]. All images were captured on the same day by the same experimenter to maintain the uniform settings adjusted at the beginning of capturing.

### 4.10. Statistical Analysis

Comparisons among the mean values obtained from different experimental groups as well as other data acquired from spectrometric, biochemical, neurochemical, and behavioral methods were subjected to one-way ANOVA analysis. The effect of each challenge compared with the vehicle-treated group was further analyzed using the Bonferroni *post hoc* test. The statistical difference was considered significant if *p* < 0.05.

## 5. Conclusions

In summary, the present study has provided for the first time that by the use of quantitative morphological, spectrometric, and functional approaches, application of OS through LEO inhalation would effectively relieve the neurochemical, biochemical and behavioral phenotypes of visceral pain. Although the detailed mechanisms participating in the beneficial effects of LEO on improving the neuronal and behavioral impairments following visceral pain are still not fully understood, effectively depressing the neuronal excitability and microglial-mediated neuroinflammation in the CeA may serve as the underlying mechanism involved in this efficacy. Considering that current options utilized for the treatment of visceral pain still exist some limiting factors, such as delay to onset or having high sedative potential [73], application of OS may be worthy of trial for clinical use as an alternative and safe strategy to counteract the visceral pain-related deficiency.

## Figures and Tables

**Figure 1 molecules-27-07659-f001:**
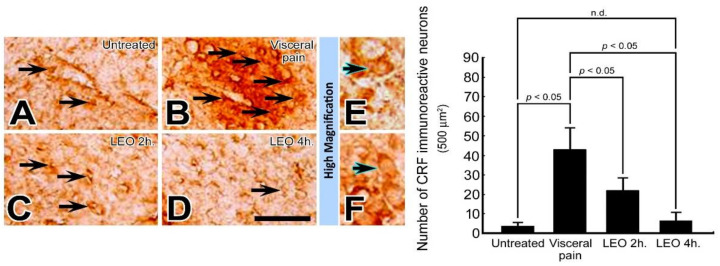
Photomicrographs (**A**–**F**) and histogram show the CRF immuno-expression in the CeA of the untreated (**A**), visceral pain (**B**), and OS groups receiving LEO for 2 h (**C**) and 4 h (**D**). Note that in the untreated group, nearly no or only a few CRF immuno-reactive neurons were detected in the CeA (**A**). However, following visceral pain, numerous CRF immuno-reactive neurons with strong staining intensity were detected in this nucleus (arrows, **B**). Nevertheless, in rats subjected to visceral pain and who received LEO inhalation, the CRF immuno-expression was significantly reduced in which only a small number of CeA neurons were stained with CRF immunoreactivity (arrows, **C**,**D**). High magnification (**E**,**F**) clearly illustrates the morphological profile of the CRF immuno-reactive neurons (blue arrows) in both untreated (**E**) and OS group (LEO 4 h.) (**F**). Quantitative evaluation paralleled well with immunohistochemical findings in which LEO inhalation successfully reduced the number of CRF immuno-reactive neurons in the CeA following the induction of visceral pain. n.d. = no significant difference. Scale bar = 25 μm in (**A**–**D**), and = 12.5 μm in (**E**,**F**).

**Figure 2 molecules-27-07659-f002:**
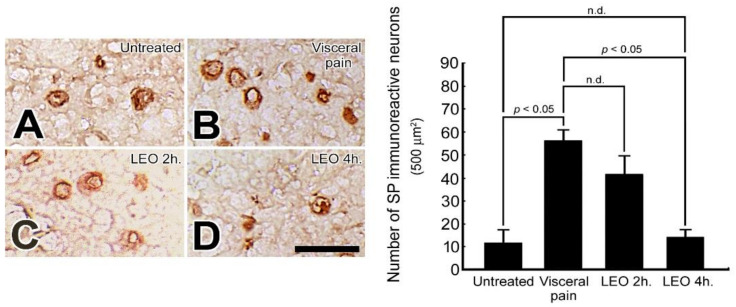
Photomicrographs (**A**–**D**) and histogram show the SP immuno-expression in the CeA of the untreated (**A**), visceral pain (**B**), and OS groups receiving LEO for 2 h (**C**) and 4 h (**D**). Note that in the untreated group, a few SP immuno-reactive neurons with mild staining intensity were detected in the CeA (**A**). However, following visceral pain, a variety of CeA neurons were positively stained with SP immunoreactivity (**B**). Nevertheless, in rats subjected to visceral pain and who received LEO inhalation, the number of SP immuno-reactive neurons was greatly decreased with the higher change detected in the group received LEO for 4 h. (**D**). Quantitative evaluation coincided well with immunohistochemical findings in which LEO inhalation successfully depressed the SP immuno-expression in the CeA following the induction of visceral pain. n.d. = no significant difference. Scale bar = 12.5 μm.

**Figure 3 molecules-27-07659-f003:**
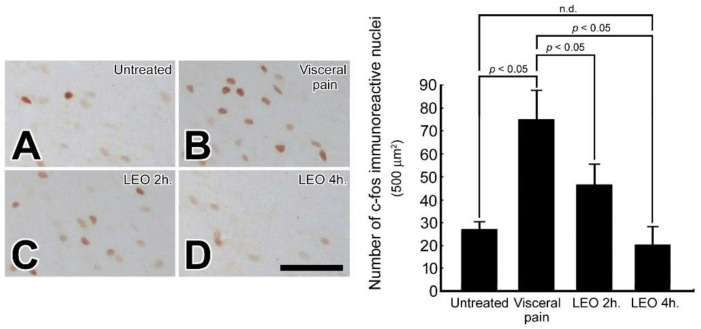
Photomicrographs (**A**–**D**) and histogram show the c-fos immuno-expression in the CeA of the untreated (**A**), visceral pain (**B**), and OS groups receiving LEO for 2 h (**C**) and 4 h (**D**). Note that in the untreated group, only a small number of c-fos immuno-reactive nuclei was detected in the CeA (**A**). However, following visceral pain, a large number of nuclei in the CeA were positive stained for c-fos immunoreactivity (**B**). Nevertheless, in rats subjected to visceral pain and who received LEO, the expression of c-fos was drastically reduced in which only a few nuclei was positively stained for c-fos (**C**,**D**). Quantitative evaluation corresponded well with immunohistochemical findings in which LEO inhalation successfully reduced the number of c-fos immuno-expression in the CeA following the induction of visceral pain. n.d. = no significant difference. Scale bar = 25 μm.

**Figure 4 molecules-27-07659-f004:**
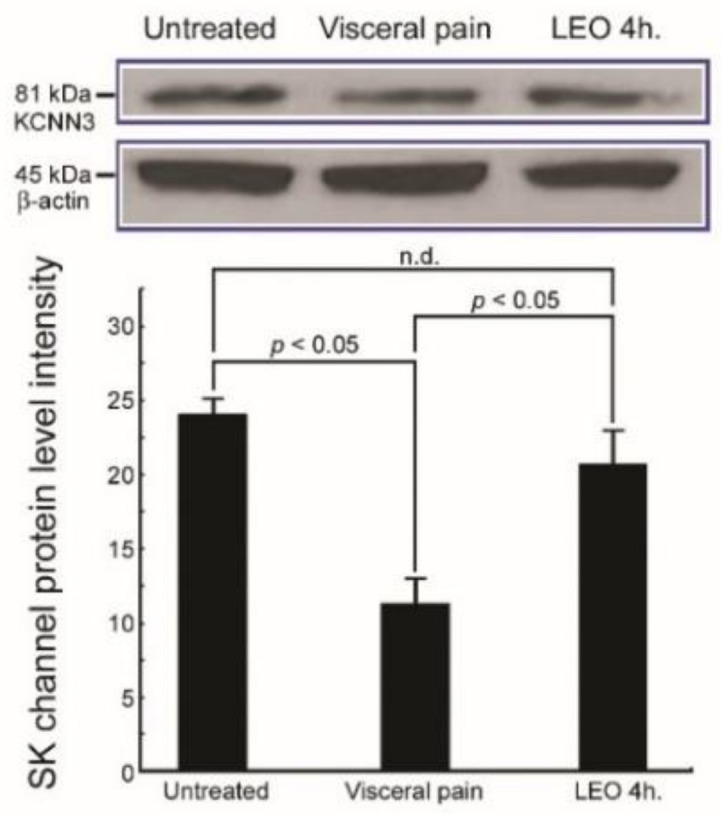
Immunoblots and histogram show the small-conductance calcium-activated potassium channel (SK channel) protein expression in the CeA of the untreated, visceral pain, and OS with LEO for 4h groups. Note that the protein level of the SK channel in the CeA was drastically decreased in response to visceral pain. However, in rats subjected to visceral pain and who received LEO inhalation, significant increase of the SK channel protein level was clearly detected in the CeA. Also note the results of anti-β-actin, which demonstrate the equal loading of proteins. n.d. = no significant difference.

**Figure 5 molecules-27-07659-f005:**
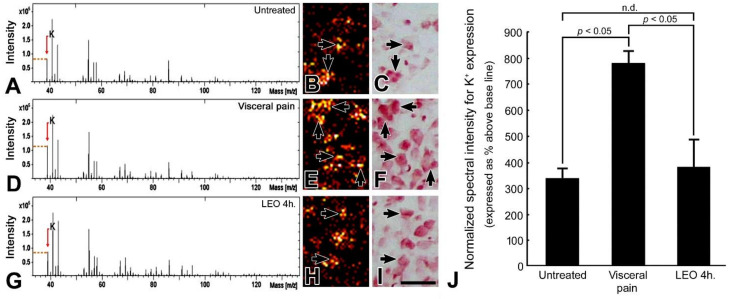
TOF-SIMS-positive spectra (**A**,**D**,**G**), ionic imaging (**B**,**E**,**H**), corresponding histological staining with H&E (**C**,**F**,**I**), and histogram (**J**) show the K^+^ expression in the CeA of the untreated (**A**–**C**), visceral pain (**D**–**F**), and OS with LEO for 4 h (**G**–**I**) groups. Note that in untreated group, only moderate K^+^ intensity (**A**) with significant intracellular localization (arrows, **B**) was detected in the CeA. However, following the induction of visceral pain, the amygdaloid K^+^ expression was drastically increased in both terms of spectral intensity (**D**) and ionic imaging (**E**). Nevertheless, in rats subjected to visceral pain and treated with LEO, the amygdaloid K^+^ expression was successfully returned to nearly normal levels in which only a mild level of K^+^ signal was detected in the CeA (**G**,**H**). Arrows in (**B**,**E**,**H**) indicates the intracellular localization of the K^+^ signal as verified by the H&E staining in (**C**,**F**,**I**). Data from normalized spectral intensity analysis corresponded well with imaging findings in which OS effectively decreased the CeA K^+^ intensity (**J**). n.d. = no significant difference. Scale bar = 50 μm in ionic and histological imaging.

**Figure 6 molecules-27-07659-f006:**
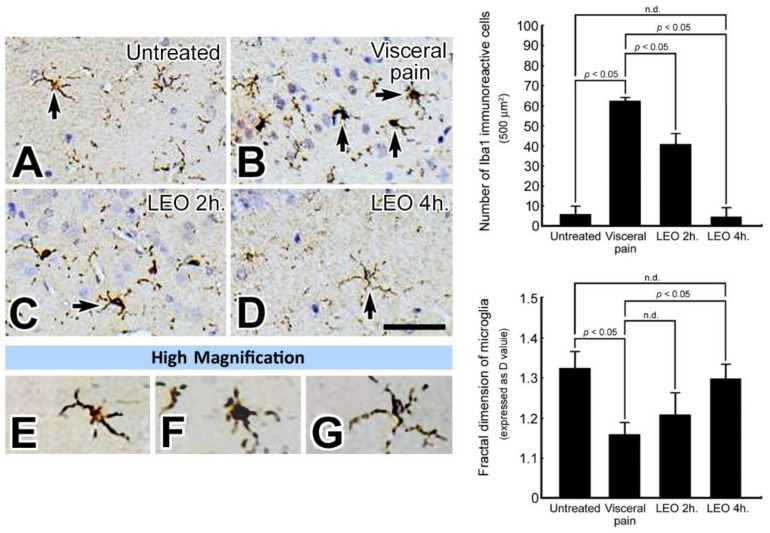
Photomicrographs (**A**–**G**) and histogram show the expression pattern of microglia (as determined by Iba1 immunohistochemistry) in the CeA of the untreated (**A**), visceral pain (**B**), and OS groups who received LEO for 2 h (**C**) and 4 h (**D**). Note that in the untreated group, only a small number of Iba1 immuno-reactive microglia with several branches and small cell bodies was detected in the CeA (arrow, **A**). However, following visceral pain, a large number of microglia with retracted branching and large cell bodies was detected in the CeA (arrows, **B**). Nevertheless, in rats subjected to visceral pain and who received LEO for 4 h, the expression pattern of microglia was much similar to that of untreated ones (arrow, **D**). High magnification (**E**–**G**) clearly demonstrates the functional status of the microglia in which a resting state was found in untreated (**E**) and OS group (**G**) while an activated state was detected in the group that suffered from visceral pain (**F**). Quantitative evaluation corresponded well with immunohistochemical findings in which OS successfully restored the branching pattern of microglia (as expressed by D value) following visceral pain. n.d. = no significant difference. Scale bar = 20 μm in (**A**–**D**), and = 10 μm in (**E**–**G**).

**Figure 7 molecules-27-07659-f007:**
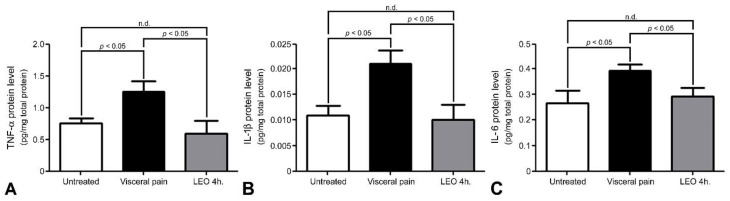
Histograms show the pro-inflammatory cytokine levels in the CeA of the untreated, visceral pain, and OS groups receiving LEO for 4 h. Note that visceral pain significantly increased TNF-α (**A**), IL-1β (**B**), and IL-6 (**C**) concentrations in the CeA as compared to the untreated group. Also note that in animals suffered from visceral pain and who received LEO, the level of pro-inflammatory cytokines was successfully reduced in which no significant difference was detected between untreated and OS groups. n.d. = no significant difference.

**Figure 8 molecules-27-07659-f008:**
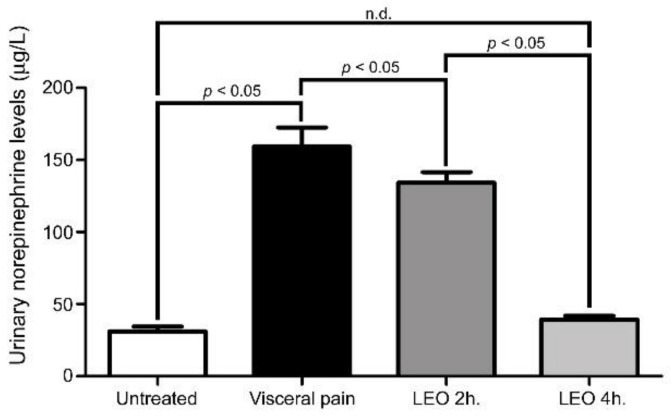
Histogram shows the urinary level of NE obtained from the untreated, visceral pain, and OS groups receiving LEO for 2 h and 4 h. Note that the urinary level of NE was drastically increased following the induction of visceral pain. However, in the animals subjected to visceral pain and receiving LEO, the urinary level of NE was effectively decreased to the value similar to that of untreated group. n.d. = no significant difference.

**Figure 9 molecules-27-07659-f009:**
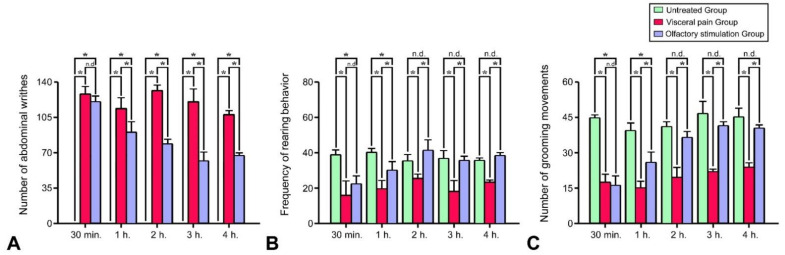
Histograms show the behavioral performances of the animals obtained from the untreated, visceral pain, and OS groups at several time points. Note that significant behavioral changes including the incidence of abdominal writhing (**A**) together with decreased rearing (**B**) and grooming movements (**C**) were detected in the animals subjected to visceral pain. However, in rats subjected to visceral pain and receiving OS for 1 h, both the incidence of abdominal writhing and the frequency of rearing and grooming behaviors were observed to be effectively improved. Also note that in animals subjected to visceral pain and received OS for 4 h, the behavioral performances were much better than that treated with OS for 1 to 3 h. * *p* < 0.05, n.d. = no significant difference.

## Data Availability

The datasets used and/or analysed during the current study are available from the corresponding author on reasonable request.

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
