# Peer review of "Olfactory Stimulation Successfully Modulates the Neurochemical, Biochemical and Behavioral Phenotypes of the Visceral Pain"

_molecules, 2022, doi:10.3390/molecules27217659_

Round 1

Reviewer 1 Report

The study of Liao et al., tested if inhaled lavender essential oil modulates the activation of the central nucleus amygdala by olfactory stimulation reducing thus the visceral pain induced by the administration of acetic acid in rats. The approach was largely biomolecular in nature, with only moderate analysis and discussion of the behavioral results underlying the effects of LEO.  The findings of this study are compelling and interesting.  The data appear of good quality and the paper is well written. However, there are several concerns and suggestions for improvement that are noted below.

1) The major concern is the lack of description and discussion about the visceral pain model used, i.e. the acetic-acid induced visceral pain (writhing test). In spite of this model is widely considered and used as a visceral pain model,   it is not viscero-specific neither a pure visceral pain model, because combine visceral and somatic mechanisms of peritoneal pain (for instance see, Le Bars et al., (2001). Animal models of nociception. Pharmacological reviews, 53(4), 597–652.; or Laird, J. et al., (2001). A new model of visceral pain and referred hyperalgesia in the mouse. Pain, 92(3), 335–342). Therefore, since the core of this study is to evaluate the actions of olfactory stimulation on the visceral pain, it should be included this limitations in the introduction and also discuss how can be influenced the observed results considering a potential painful somatic activation on the amygdala.

2) In this study there is not a graphical representation of the time-course of the pain behaviors (writhing responses). Additionally to figure 9A,  I think that the inclusion of a comparison between the time-course of the inhaled water vapors and LEO groups treated with acetic acid could help to get an overview about the evolution of the  behavioral pain responses and the duration of the LEO actions as well as their correlation with all the neurochemical and biochemical markers tested. Since the responses were video recorded, the authors can make a graph considering several time points during the 4 h.

3) What is the rationale to use males? Could be the effect of LEO different on the pain responses if the rats had been female? Also the age of the animals could be an important factor. There is evidence about these facts that must be considered to mention in the manuscript as it has been reported (Premachandran et al., Sex Differences in the Development of the Rodent Corticolimbic System. Front Neurosci. 2020 Sep 30;14:583477.; Murphy et al.,. Sex differences in the activation of the spinoparabrachial circuit by visceral pain. Physiol Behav. 2009 May 25;97(2):205-12. Wang et al.,Sex differences in functional brain activation during noxious visceral stimulation in rats. Pain. 2009 Sep;145(1-2):120-128).

Minor points:

It is not clear the criteria and how the experimental animals were selected and distributed for some studies. In the section 4.3 and 4.6 is mentioning that half amount of the total animals were used  for each type of studies. Were these animals randomly selected or based in their behavioral responses or another criteria?. For the experiments in sections 4.7, 4.8 and 4.9, which animals were used and in what number?

Line 67: I guess that should be “massage” instead  “message”

Author Response

We deeply appreciate the valuable and constructive comments from this Reviewer in which all necessary amendments were made to meet the suggestions for this revised manuscript as follows:

1. We agree with this Reviewer’s valuable comments that the model used in this study is not a pure visceral pain model since intraperitoneal injection of algogenic agent may also stimulate the parietal peritoneum that generally belongs to the somatic structure. In this regard, we have briefly pointed out this limitation in the Introduction section, and further discuss how
this restriction can influence, and/or interference with the current results (Please see p. 3, lns.
101-109, and p. 11, lns. 547-558).

2. We thank the constructive suggestions from this Reviewer in which inclusion a
graphical data depicting the behavioral changes following visceral pain at several time points would not only outlining an overview about the evolution of visceral pain-induced behavioral responses and the action of LEO, but also illustrating the temporal correlation between behavioral profiles with all neurochemical and biochemical markers tested. With regard to this viewpoint, we have replaced the Fig. 9 by adding new data to completely show the time
course alterations of writhing behavior in response to visceral pain, and to further clarify the chronological development of antinociceptive function of LEO exerted by current strategy (Please see p. 9, lns. 428-436, and the revised Fig. 9).

3. We deeply appreciate the valuable comments from this Reviewer that the potential impacts of both gender and age on the neuronal processing of visceral pain should not be overlooked. As a result, we have addressed this issue in the revised manuscript, and highlighted the importance of evaluating these two factors on the modulation of visceral pain in the future studies (Please see p. 11, lns. 559-570).

For Minor points:

1. The criteria and selection (as well as the number) of animals used in each type of study have now been clearly described in the revised version of the manuscript (Please see p. 12, lns. 613-615, p. 13, lns. 659-660, 661-665, and p. 14, lns. 677-679, 689-691, 694-695).

2. We thank the kind notification of this Reviewer in which the mistyped word
“message” has been corrected to “massage” in the revised manuscript (Please see p. 2, ln. 68).

Reviewer 2 Report

The authors examined the effects of olfactory stimulation with lavender essential oil (LEO) on neurochemical, biochemical and behavioral changes induced by visceral pain, and  demonstrated that it suppressed the enhanced neuronal excitability and neuroinflammation in mice. Although the findings suggest the relieving effect of LEO on visceral pain-induced pathological changes and may be of interest to the researchers in the pain and alternative medicine research fields, I have some concerns described below.

1) The number of substance P immune-positive neurons differs between text (line 151) and Fig. 2 (graph). Similarly, the K+ signal is different between text (lines 257-258) and Fig. 5 (graph). These seriously diminishes the credibility of the study.

2) Photos at higher magnification should also be shown to better illustrate the morphology of CRF-positive neurons (Fig. 1) and resting/activated microglia (Fig. 6).

3) For SK channels, the authors should use the term such as "SK channel protein level" rather than "SK channel activity" because they only examine its protein levels by Western blotting and not the function of this channel.

4) The authors state that the beneficial effects of LEO in relieving visceral pain are time-dependent, as the maximum effect was detected at 4 hours following the exposure of LEO. However, they only examined the effects of LEO at two time points (2 and 4 hours after LEO exposure). Thus, it is not possible to determine whether the effect is maximal at 4 hours.

5) Ref. 64 (line 661) is not in the reference list.

Author Response

We thank the valuable and constructive comments from this Reviewer in which all necessary amendments were made to meet the requirements for this revised manuscript as follows:

1. We thank this Reviewer’s kind notification in which the number of substance P
immuno-reactive neurons as well as the normalized spectral intensity of the K+
signal has been carefully checked and corrected to match the data shown in the corresponding graphs (Please see p. 4, lns. 163-164, and p. 6, lns. 274-276.)

2. The higher magnification of the photomicrographs has been provided to better illustrate the morphology of the CRF immuno-positive neurons and the resting/activated stage of the microglia (Please see p. 3, lns. 144-145 in the revised Fig. 1, and p. 7, lns. 349-351 in the revised Fig. 6).

3. We thank the valuable suggestion from this Reviewer in which the term “SK channel protein level” has now been used to describe the result collected from SK channel immunoblotting in the revised manuscript (Please see p. 1, lns. 29-30, p. 5, lns. 228, 230, 232, 234-235, 256, 258-259, and p. 13, ln. 658).

4. We agree with this Reviewer that it may not be appropriate to state that the “maximal” effect of olfactory stimulation (OS) was detected at 4 hours following the exposure of LEO, since the present study did not test the potential effects of OS at longer time points. In this regard, we have rephrased the sentence to state that “the better effects of OS were detected at 4 hours following the exposure of LEO” to properly fit the experimental design and the current results (Please see p. 9, lns. 447-448).

5. The original reference [64] has now been added as reference [73] in the revised manuscript (Please see the Reference section). 

Round 2

Reviewer 1 Report

In the present version of the manuscript all my concerns have been resolved.

Reviewer 2 Report

No more comments.